# Telomere length is associated with growth in children in rural Bangladesh

**Audrie Lin[1]\*, Andrew N Mertens[1], Benjamin F Arnold[2], Sophia Tan[1], Jue Lin[3], Christine P Stewart[4], Alan E Hubbard[1], Shahjahan Ali[5], Jade Benjamin-Chung[6], Abul K Shoab[5], Md Ziaur Rahman[5], Syeda L Famida[5], Md Saheen Hossen[5], Palash Mutsuddi[5], Salma Akther[5], Mahbubur Rahman[5], Leanne Unicomb[5], Ruchira Tabassum Naved[7], Md Mahfuz Al Mamun[7], Kausar Parvin[7], Firdaus S Dhabhar[8], Patricia Kariger[9], Lia CH Fernald[9], Stephen P Luby[10], John M Colford Jr[1]**

[1]Division of Epidemiology and Biostatistics, School of Public Health, University of California, Berkeley, Berkeley, United States; [2]Francis I. Proctor Foundation, University of California San Francisco, San Francisco, United States; [3]Department of Biochemistry and Biophysics, University of California, San Francisco, San Francisco, United States; [4]Department of Nutrition, University of California, Davis, Davis, United States; [5]Infectious Disease Division, International Centre for Diarrhoeal Disease Research, Dhaka, Bangladesh; [6]Department of Epidemiology & Population Health, Stanford University, Stanford, United States; [7]Health System and Population Studies Division, International Centre for Diarrhoeal Disease Research, Dhaka, Bangladesh; [8]Department of Psychiatry & Behavioral Sciences, Sylvester Comprehensive Cancer Center, Miller School of Medicine, University of Miami, Miami, United States; [9]Division of Community Health Sciences, School of Public Health, University of California, Berkeley, Berkeley, United States; [10]Division of Infectious Diseases and Geographic Medicine, Stanford University, Stanford, United States

**\*For correspondence:**
audrielin@berkeley.edu

## Abstract

**Background:** Previously, we demonstrated that a water, sanitation, handwashing, and nutritional intervention improved linear growth and was unexpectedly associated with shortened childhood telomere length (TL) (Lin et al., 2017). Here, we assessed the association between TL and growth.

**Methods:** We measured relative TL in whole blood from 713 children. We reported differences between the 10th percentile and 90th percentile of TL or change in TL distribution using generalized additive models, adjusted for potential confounders.

**Results:** In cross-sectional analyses, long TL was associated with a higher length-for-age Z score at age 1 year (0.23 SD adjusted difference in length-for-age Z score [95% CI 0.05, 0.42; FDR-corrected p-value = 0.01]). TL was not associated with other outcomes.

**Conclusions:** Consistent with the *metabolic telomere attrition hypothesis*, our previous trial findings support an adaptive role for telomere attrition, whereby active TL regulation is employed as a strategy to address 'emergency states' with increased energy requirements such as rapid growth during the first year of life. Although short periods of active telomere attrition may be essential to promote growth, this study suggests that a longer overall initial TL setting in the first 2 years of life could signal increased resilience against future telomere erosion events and healthy growth trajectories.

**Funding:** Funded by the Bill and Melinda Gates Foundation.

**Clinical trial number:** NCT01590095

## Introduction

Children living in low-income countries are often exposed to a broad array of environmental insults leading to malnutrition, impaired development, and early mortality (*Black et al., 2017*; *Victora et al., 2008*; *Wang et al., 2016*). Early life is a period of heightened developmental plasticity and sensitivity to environmental conditions (*Barker, 2007*). Under the developmental origins of health and disease framework, environmental stimuli alter foetal and early life programming to shape physiology and contribute to adult disease (*Barker, 2007*; *Wadhwa et al., 2009*).

Growing evidence implicates telomere length (TL) attrition as a potentially critical underlying mechanism that links early life adverse events with poor health outcomes later in life (*Price et al., 2013*). The erosion of telomeres, the DNA repeats, and protein complexes at the ends of chromosomes, during cell division gradually leads to cell senescence (*Blackburn, 2001*). Shorter TL is associated with diabetes, heart disease, and early mortality (*Cawthon et al., 2003*; *Fitzpatrick et al., 2007*; *Salpea et al., 2010*). In our previous study, we hypothesized that early life interventions designed to improve nutrition and decrease environmental faecal contamination would reduce infections and inflammation – exposures associated with TL attrition – and, thereby, slow TL attrition (*Lin et al., 2017*). The interventions improved linear growth (length-for-age Z scores) but were unexpectedly associated with shortened TL during the first year of life (*Lin et al., 2017*; *Luby et al., 2018*), findings that challenged the prevailing paradigm that early-life stressors shorten TL (*Ridout et al., 2015*) and motivated the hypothesis of the present study: that accelerated TL attrition in early life could be associated with improved growth.

Growth and neurodevelopment occur because of hypertrophy and cell proliferation in the growth plate and brain (*Laron, 2009*; *Loveridge and Noble, 1994*), and TL shortens in proportion to the number of cell replications. Measurements of growth and TL may be related or unrelated epiphenomena of an underlying causal process linking environmental insults to child neurodevelopmental outcomes. The pathogeneses of common phenotypes of malnutrition are complex and poorly understood. Stunting (low length for age) reflects chronic exposure to undernutrition and infections, wasting (low weight for length) indicates acute weight loss, and underweight (low weight for age) serves as a composite indicator for wasting, stunting, or both (*World Health Organization, 2010*). Stunted, wasted, and underweight children often experience impaired neurodevelopment and educational performance, increased risk of infections and chronic disease, reduced adult economic productivity, and increased mortality risk (*World Health Organization, 2010*). The rapidly expanding field of infant telomere biology may contribute to our understanding of the causal pathway leading to impaired neurodevelopment because the highest rates of TL attrition, growth, and development occur during the first 2 years of life (*Elwood, 2004*). During this dynamic period, increases and decreases in TL among individual children have been observed (*Bosquet Enlow et al., 2020*). Here, we evaluated the potential association between TL and growth in early life.

## Materials and methods

### Study design

The WASH Benefits study was a cluster-randomised controlled trial designed to study the effects of improved drinking water, sanitation, handwashing, and nutrition on child growth and diarrhoea (*Luby et al., 2018*). The study was conducted in rural villages in the Gazipur, Mymensingh, Tangail, and Kishoreganj districts of Bangladesh. This substudy only focused on children in the control and the combined nutrition, water, sanitation, and handwashing (N + WSH) intervention arms. Clusters were defined as eight neighbouring households with eligible pregnant women.

### Participants

We enrolled pregnant women in their first two trimesters and their in utero children between 31 May 2012 and 7 July 2013. Households with relocation plans in the following year, households without home ownership, and households with high iron content in their water sources were excluded from the study. This analysis focused on index children, defined as in utero children of enrolled women. In this substudy, children were excluded from blood collection if any two of the following criteria for moderate to severe dehydration were met: (1) restless, irritable, (2) sunken eyes, (3) drinks eagerly, thirsty, and (4) pinched skin returns to normal position slowly. Children who were listless or unable

to perform normal activities were also excluded from blood collection. Two children were excluded based on these criteria.

## Procedures

The trial consisted of six intervention arms and a double-sized control arm (*Luby et al., 2018*). This substudy only assessed children in the control arm and the combined intervention arm. The combined intervention consisted of the following components: water treatment (Aquatabs; NaDCC) and safe storage vessel, sanitation (child potties, sani-scoop hoes to remove faeces, and a double pit latrine with a hygienic water seal), handwashing (handwashing stations near the latrine and kitchen, including soapy water bottles and detergent soap), and nutrition (lipid-based nutrient supplements [Nutriset, Malaunay, France] that included ≥100 % of the recommended daily allowance of 12 vitamins and 9 minerals with 9.6 g of fat and 2.6 g of protein daily for children 6–24 months of age and age-appropriate recommendations on maternal nutrition and infant feeding practices) (*Luby et al., 2018*). To promote recommended behaviours, community health promoters visited study compounds in the intervention arm at least once per week during the initial 6 months of the trial and at least once biweekly thereafter. Participants in the control arm did not receive interventions or promoter visits.

## Relative TL measurements

Relative TL, expressed as the ratio of telomere to single-copy gene abundance (T/S ratio), was measured in whole blood at Year 1 (median age 14 months) and Year 2 (median age 28 months) after intervention delivery. The protocol for the measurement of relative TL, by quantitative polymerase chain reaction (qPCR), was previously described (*Lin et al., 2017*). We measured relative TL by quantitative polymerase chain reaction (qPCR), expressed as the ratio of telomere to single-copy gene abundance (T/S ratio) (*Cawthon, 2002*; *Lin et al., 2010*). Genomic DNA was extracted from heparin-anti-coagulated whole blood stored at –80 °C using the QIAamp DNA Mini Kit (QIAGEN, Hilden, Germany). DNA quantity and quality was assessed using a NanoDrop 2000c Spectrophotometer (Nanodrop Products, Wilmington, DE). DNA was stored at –80 °C for batch TL analysis. Of the 1384 DNA samples, 8 did not pass quality control (an OD260/OD280 between 1.7 and 2.0 and concentration greater than 10 ng/µl) and one sample failed to amplify, resulting in 1375 samples with valid TL data. The intra-class correlation coefficients were as follows: 0.04 T/S ratio for TL at Year 1, 0.08 T/S ratio for TL at Year 2, and 0.18 T/S ratio for the change in TL between Years 1 and 2.

The telomere qPCR primers were *tel1b* [5'-CGGTTT(GTTTGG)$_5$GTT-3'], used at a final concentration of 100 nM, and *tel2b* [5'-GGCTTG(CCTTAC)$_5$CCT-3'], used at a final concentration of 900 nM. The single-copy gene (human β-globin) qPCR primers were *hbg1* [5'-GCTTCTGACACAACTG TGTTCACTAGC-3'], used at a final concentration of 300 nM, and *hbg2* [5'-CACCAACTTCATCCAC GTTCACC-3'], used at a final concentration of 700 nM. The final reaction mix consisted of the following: 20 mM Tris–hydrochloride, pH 8.4; 50 mM potassium chloride; 200 µM each deoxyribonucleotide triphosphate; 1 % dimethyl sulfoxide; 0.4× SYBR green I; 22 ng *Escherichia coli* DNA; 0.4 Units of platinum Taq DNA polymerase (Invitrogen Inc, Carlsbad, CA), and approximately 6.6 ng of genomic DNA per 11 µl reaction.

A threefold serial dilution of a commercial human genomic DNA (Sigma-Aldrich, cat#11691112001) containing 26, 8.75, 2.9, 0.97, 0.324, and 0.108 ng of DNA was included in each PCR run as the reference standard. The quantity of targeted templates in each sample was determined relative to the reference DNA sample by the maximum second-derivative method in the Roche LC480 program. The reaction was carried out in a Roche LightCycler 480 in 384-well plates, with triplicate wells for each sample. Dixon Q test was used to exclude outliers from the triplicates. The average of the T and S triplicate wells after outlier removal was used to calculate the T/S ratio for each sample. The same reference DNA was used for all PCR runs.

We applied a telomere (T) thermal profile consisting of denaturing at 96 °C for 1 min followed by 30 cycles of denaturing at 96 °C for 1 s and annealing or extension at 54 °C for 60 s with fluorescence data collection and a single-copy gene (S) thermal profile consisting of denaturing at 96 °C for 1 min followed by eight cycles of denaturing at 95 °C for 15 s, annealing at 58 °C for 1 s, and extension at 72 °C for 20 s, followed by 35 cycles of denaturing at 96 °C for 1 s, annealing at 58 °C for 1 s, extension at 72 °C for 20 s, and holding at 83 °C for 5 s with data collection. The T/S ratio for each sample was

measured in duplicate runs, each with triplicate wells. When the duplicate T/S values disagreed by more than 7%, the sample was run in triplicate and the two closest values were used.

Eight control genomic DNA samples were included to calculate a normalising factor for each run. In each batch, the T/S ratio of each control DNA was divided by the average T/S ratio for the same DNA from 10 runs to generate a normalising factor that was then used to correct the participant DNA samples to generate the final T/S ratio. The DNA extraction and TL measurements were performed in two batches (3.5 months apart) using the same lots of reagents. To account for assay batch variations, 48 samples from the first batch were re-assayed together with the second batch of samples; then, data from the second batch of samples were adjusted by a factor of 1.05 (derived from the systematic difference between the first batch values versus the second batch values for these 48 samples).

## Anthropometric measurements

Following standard protocols for anthropometric outcomes measurement (*Cogill,, 2003*; *de Onis et al., 2004*), pairs of trained anthropometrists measured recumbent length (accurate to 0.1 cm), weight without clothing, and head circumference in triplicate. We used the median of the three measurements to calculate length-for-age, weight-for-age, weight-for-length, and head circumference-for-age Z scores standardised to the WHO 2006 child growth standards using publicly available software (https://www.who.int/tools/child-growth-standards/software). The WHO 2006 multicentre growth reference study constructed Z score curves for boys and girls aged 0–60 months based on measurements from a sample of healthy breastfed infants and young children living in the United States, Oman, Norway, Brazil, Ghana, and India (*de Onis et al., 2006*). Z score calculations (Z score = (raw measurement – reference population mean)/reference population standard deviation) in this study used reference population means and standard deviations derived from the WHO 2006 child growth standards. Child age was determined using birthdates verified when possible using vaccination cards. Age at the time of measurement was compared against the Z score curves for the WHO 2006 reference population. Length, weight, and head circumference were measured at Years 1 and 2 post-intervention (median ages 14 and 28 months). We excluded children from Z score analyses if their growth measurements were outside biologically plausible ranges according to WHO recommendations (*de Onis et al., 2004*). Due to widespread malnutrition in low- and middle-income countries, the mean anthropometric Z scores will generally be <0 during the first 2 years of life (*Black et al., 2013*). Stunting was defined as length-for-age Z scores below –2 standard deviations from the WHO length-for-age standards median, underweight was defined as weight-for-age Z scores < –2, and wasting was defined as weight-for-length Z scores < –2.

## Exposures

We assessed the following exposures: TLs at Year 1, TLs at Year 2, and change in TL between Years 1 and 2 post-intervention.

## Outcomes

We assessed child length-for-age, weight-for-age, weight-for-length, and head circumference-for-age Z scores at Year 1 and Year 2. We measured the change in child length-for-age, weight-for-age, weight-for-length, and head circumference-for-age Z scores from Year 1 to Year 2 post-intervention. Because anthropometric Z scores reflect attained growth, we also assessed child weight velocity (in kg/month), length velocity (in cm/month), and head circumference velocity (in cm/month) from Year 1 to Year 2 to assess the growth process.

## Statistical analysis

The pre-registered analysis protocol, data, and code for the substudy are available (https://osf.io/9snat/). Analyses were conducted using R statistical software version 4.0.3.

The substudy was nested within the environmental enteric dysfunction substudy (*Lin et al., 2019*). Because the substudy consisted of the control and combined intervention arms only, we enrolled 713 children from 135 clusters with an average of 5 children per cluster. To estimate the minimum detectable effect between quartiles of TL, we assumed a two-sided alpha of 5 % and standard deviations of +1.00 LAZ, + 0.97 WAZ, and +0.90 WLZ. The cluster-level intra-class correlation coefficient within our study was 0.07 for LAZ, 0.04 for WAZ, and 0.02 for WLZ. The trial had 80 % power to detect a +

0.32 difference in LAZ, a + 0.31 difference in WAZ, and a + 0.28 difference in WLZ between quartiles of TL.

For each exposure-outcome pair, we conducted exploratory data analyses that plotted the relationship between telomere exposures and growth outcomes and summarised the patterns between them using cubic splines, with the bandwidth chosen using generalised cross-validation (*Wood et al., 2017*). We estimated Bayesian 95 % simultaneous confidence intervals around the fitted curves (*Nychka, 1988*). We tested for the bivariate association between the exposure and outcome using a permutation test with Spearman's rank correlation test statistic to determine if it differed from zero.

We summarised mean anthropometric Z scores (length-for-age, weight-for-age, weight-for-length, and head circumference-for-age Z scores), change in anthropometric Z scores, length velocity, weight velocity, and head circumference velocity across the distribution of TL or change in TL using natural smoothing splines (generalised additive models), both unadjusted and adjusted for potential confounding covariates. We estimated differences in Z scores and pointwise confidence intervals compared to a reference level of the lowest observed TL or change in TL. We reported the differences and confidence intervals between the 10th percentile and 90th percentile of TL or change in TL distribution using predictions from the generalised additive models.

We pre-screened each covariate separately to assess whether they were associated with each outcome prior to including them in the model. We used the likelihood ratio test to assess the association between each outcome and each covariate and included covariates with a *P*-value < 0.20 in the analysis. We excluded categorical covariates that had little variation in the study population (prevalence < 5%).

*Supplementary file 1a* includes the full list of pre-specified covariates tested for inclusion in adjusted models. Briefly, the list includes covariates pertaining to the child (e.g., age, sex, and prior growth measurements), mother (e.g., age, height, education, depression, perceived stress, and exposure to physical, sexual, or emotional intimate partner violence), and household (e.g., food insecurity, assets, and treatment group). We reported unadjusted p-values and adjusted for multiple testing by controlling the false discovery rate (at an FDR of 5%) within each hypothesis using the Benjamini–Hochberg procedure.

## Results

### Child characteristics

TL and anthropometry measurements were available from 662 children at Year 1 and 713 children at Year 2 (*Figure 1*); 557 children had measurements in both years. The median age of the children was 14.3 (IQR: 12.6–15.6) months at Year 1 and 28.2 (IQR: 26.9–29.6) months at Year 2 (*Table 1*). After 1 year post-enrolment, 27 % of children were stunted, 24 % were underweight, and 12% were wasted. The median length-for-age Z score, weight-for-age Z score, weight-for-length Z score, and head circumference-for-age Z score was −1.41,−1.30, –0.89, and –1.81, respectively (*Table 1*). At Year 2, the stunting, underweight, and wasting prevalence remained stable. The median length-for-age Z score, weight-for-age Z score, weight-for-length Z score, and head circumference-for-age Z score was −1.54, −1.55, −1.00, and –1.78. Diarrhoea prevalence was higher at the first visit (14%, median age 14 months) compared to the second visit (8%, median age 28 months).

### Maternal characteristics

At enrolment, the mean (± SD) age of the women was 24 (±5) years, with a median height of 151 cm (IQR: 147–154) (*Table 1*). Women completed a median of 7 years of schooling (IQR: 4–9). Fifty-six percent of women reported experiencing intimate partner violence in their lifetime. The median CESD-R score of 12 (IQR: 9–17) at Years 1 and 2 was below the cut point for clinical depression (score of 16); 28 % of women at Year 1 and 25% at Year 2 had a score at or above 16, indicating depressive symptomatology. The median perceived stress scale score was 14 (IQR: 10–18), similar to the reference norm in the United States (*Cohen, 1994*).

### TL at Year 1 and growth

At Year 1, the median TL was 1.42 T/S ratio (IQR: 1.28–1.56) (*Table 1*). At Year 1, longer TL was concurrently associated with taller children (+0.23 SD adjusted difference in length-for-age Z score between

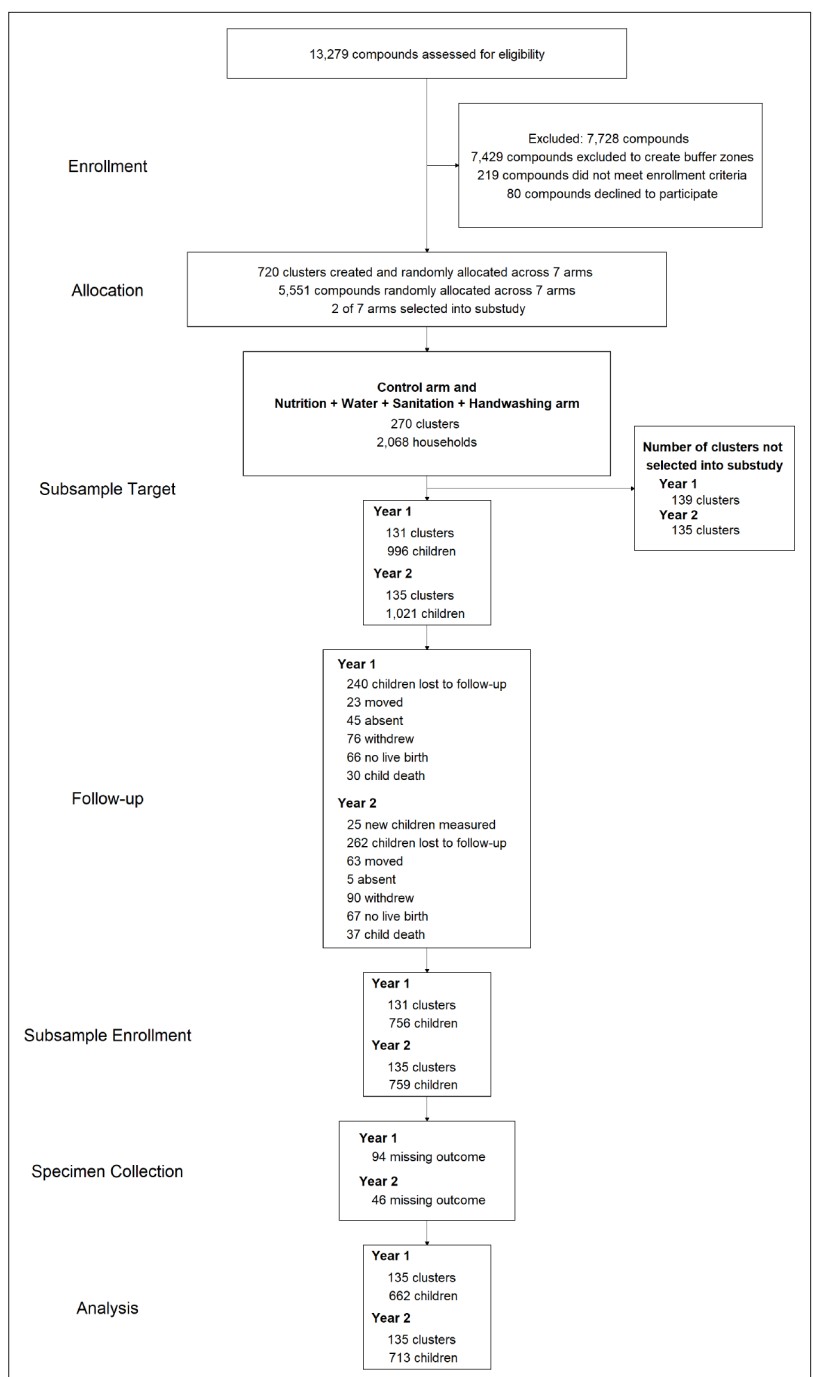

**Figure 1.** Diagram of participants at each phase of the telomere length and growth substudy within the WASH Benefits trial.

The online version of this article includes the following source code for figure 1:

**Source code 1.** Source code for *Figure 1*.

the 10th and 90th percentile [95% CI 0.05, 0.42; FDR-corrected p-value = 0.01]) (*Table 2* and *Supplementary file 1b*; *Figure 2*, *Figure 2—figure supplement 1*). TL at Year 1 was unassociated with concurrent or subsequent weight-for-age, weight-for-length, or head circumference-for-age Z score (*Table 2* and *Supplementary file 1b*; *Figure 2*, *Figure 2—figure supplement 1*). Furthermore, TL at Year 1 was unassociated with growth velocities or change in length-for-age Z score, weight-for-age,

**Table 1.** Characteristics of participants.

| | | N (%) or median (IQR) |
|---|---|---|
| Child | Female (%) | 417 (51%) |
| | Age (months) at Year 1 | 14.3 (12.6, 15.6) |
| | Age (months) at Year 2 | 28.2 (26.9, 29.6) |
| | Months between telomere length measurements at Year 1 and Year 2 | 13.9 (13.5, 14.7) |
| Telomere length at Year 1 | T/S ratio* | 1.42 (1.28, 1.56) |
| Telomere length at Year 2 | T/S ratio* | 1.43 (1.29, 1.58) |
| Change in telomere length between Year 1 and Year 2 | T/S ratio* | 0.04 (-0.22, 0.25) |
| Anthropometry (age 3 months, Month 3) | Length-for-age Z score | −1.28 (−1.99, −0.53) |
| | Weight-for-age Z score | −1.18 (−1.84, −0.50) |
| | Weight-for-length Z score | −0.26 (−1.11, 0.44) |
| | Head circumference-for-age Z score | −1.73 (−2.39, −1.01) |
| Anthropometry (age 14 months, Year 1) | Length-for-age Z score | −1.41 (−2.08, −0.77) |
| | Weight-for-age Z score | −1.30 (−1.98, −0.68) |
| | Weight-for-length Z score | −0.89 (−1.60, −0.25) |
| | Head circumference-for-age Z score | −1.81 (−2.40, −1.19) |
| Anthropometry (age 28 months, Year 2) | Length-for-age Z score | −1.54 (−2.25, −0.94) |
| | Weight-for-age Z score | −1.55 (−2.09, −0.90) |
| | Weight-for-length Z score | −1.00 (−1.59,−0.37) |
| | Head circumference-for-age Z score | −1.78 (−2.37, −1.22) |
| Diarrhoea (age 14 months, Year 1) | Caregiver-reported 7 day recall (%) | 104 (14%) |
| Diarrhoea (age 28 months, Year 2) | Caregiver-reported 7 day recall (%) | 56 (8%) |
| Mother | Age (years) | 23 (20, 27) |
| Anthropometry at enrolment | Height (cm) | 150.5 (147.1, 153.9) |
| Education | Schooling completed (years) | 7 (4, 9) |
| Depression at Year 1 | CESD-R score** | 10 (6, 16) |
| Depression at Year 2 | CESD-R score** | 10 (5, 17) |
| Perceived stress at Year 2 | Perceived Stress Scale score | 14 (10, 18) |
| Physical, sexual, or emotional intimate partner violence | Any lifetime exposure: number of women (%) | 398 (56%) |

*The unit for relative telomere length is the T/S ratio. Telomere length was measured by quantitative PCR (qPCR), a method that determines relative telomere length by measuring the factor by which each DNA sample differs from a reference DNA sample in its ratio of telomere repeat copy number (T) to single-copy gene copy number (S).

**CESD-R = Center for Epidemiologic Studies Depression Scale Revised.

The online version of this article includes the following source code for table 1:

**Source code 1.** Source code for *Table 1*.

**Table 2.** Association between telomere length at Year 1 and growth.

| Exposure | Outcome | N | 10th percentile | 90th percentile | Outcome, 90th percentile vs. 10th percentile | | | |
|---|---|---|---|---|---|---|---|---|
| | | | | | Adjusted† | | | |
| | | | | | Predicted outcome at 10th percentile | Predicted outcome at 90th percentile | Coefficient (95% CI) | p-value |
| Telomere length at Year 1 (T/S ratio) | LAZ Year 1 | 638 | 1.2 | 1.7 | −1.54 | −1.31 | 0.23 (0.05, 0.42) | 0.01* |
| | WAZ Year 1 | 638 | 1.2 | 1.7 | −1.21 | −1.20 | 0.01 (−0.2, 0.23) | 0.91 |
| | WLZ Year 1 | 636 | 1.2 | 1.7 | −0.68 | −0.83 | −0.16 (-0.36, 0.05) | 0.14 |
| | HCZ Year 1 | 638 | 1.2 | 1.7 | −1.82 | −1.81 | 0.01 (−0.18, 0.2) | 0.94 |
| | LAZ Year 2 | 542 | 1.2 | 1.7 | −1.51 | −1.45 | 0.06 (−0.04, 0.16) | 0.22 |
| | WAZ Year 2 | 565 | 1.2 | 1.7 | −1.77 | −1.70 | 0.07 (−0.03, 0.17) | 0.18 |
| | WLZ Year 2 | 568 | 1.2 | 1.7 | −0.99 | −1.01 | −0.03 (−0.14, 0.09) | 0.68 |
| | HCZ Year 2 | 565 | 1.2 | 1.7 | −1.85 | −1.81 | 0.04 (−0.18, 0.25) | 0.73 |
| | Change in LAZ between Year 1 and Year 2 | 568 | 1.2 | 1.7 | −0.23 | −0.28 | −0.05 (−0.14, 0.04) | 0.29 |
| | Change in WAZ between Year 1 and Year 2 | 572 | 1.2 | 1.7 | −0.37 | −0.38 | 0 (−0.11, 0.1) | 0.94 |
| | Change in WLZ between Year 1 and Year 2 | 565 | 1.2 | 1.7 | −0.24 | −0.20 | 0.04 (−0.09, 0.17) | 0.56 |
| | Change in HCZ between Year 1 and Year 2 | 545 | 1.2 | 1.7 | −0.16 | −0.11 | 0.06 (−0.05, 0.17) | 0.31 |
| | Length velocity between Year 1 and Year 2 | 541 | 1.2 | 1.7 | 0.80 | 0.80 | 0 (−0.02, 0.03) | 0.79 |
| | Weight velocity between Year 1 and Year 2 | 541 | 1.2 | 1.7 | 0.15 | 0.15 | 0 (−0.01, 0.01) | 0.74 |
| | Head circumference velocity between Year 1 and Year 2 | 545 | 1.2 | 1.7 | 0.15 | 0.16 | 0.01 (0, 0.02) | 0.13 |

N, 10th percentile, and 90th percentile are from the adjusted analyses.

T/S ratio = unit for relative telomere length; LAZ = length-for-age Z score; WAZ = weight-for-age Z score; WLZ = weight-for-length Z score; HCZ = head circumference-for-age Z score.

*p-value<0.05 after adjusting for multiple comparisons using the Benjamini–Hochberg procedure.

†Adjusted for pre-specified covariates: Child age, child sex, birth order, prior child length, and weight measurements (included in Year 2 outcomes only), time between anthropometry measurements (included in growth velocity and change in growth measurements between Year 1 and Year 2 outcomes only), season of measurement, caregiver-reported diarrhoea, mother's age, mother's height, mother's education level, mother's Center for Epidemiologic Studies Depression Scale Revised (CESD-R) score, mother's Perceived Stress Scale score, mother's lifetime exposure to physical, sexual, and emotional intimate partner violence, household food insecurity, number of children < 18 years in the household, number of individuals living in the compound, distance in minutes to the primary water source, household floor materials, household wall materials, household electricity, and household assets (wardrobe, table, chair, clock, khat, chouki, radio, television, refrigerator, bicycle, motorcycle, sewing machine, mobile phone, cattle, goats, and chickens), and treatment arm (control or N + WSH) (*Supplementary file 1a*).

and weight-for-length Z score between Years 1 and 2 (*Table 2* and *Supplementary file 1b*; *Figure 2*, and *Figure 2—figure supplement 1*).

## TL at Year 2 and growth

At Year 2, the median TL was 1.43 T/S ratio (IQR: 1.29–1.58) (*Table 1*). Similar to the Year 1 results, the association between TL at Year 2 and length-for-age Z score was positive (+0.08 SD adjusted difference between the 10th and 90th percentile [95% CI −0.03, 0.19]), but the association was not significant (FDR-corrected p-value≥0.05) (*Table 3* and *Supplementary file 1c*; *Figure 2*, *Figure 2—figure*

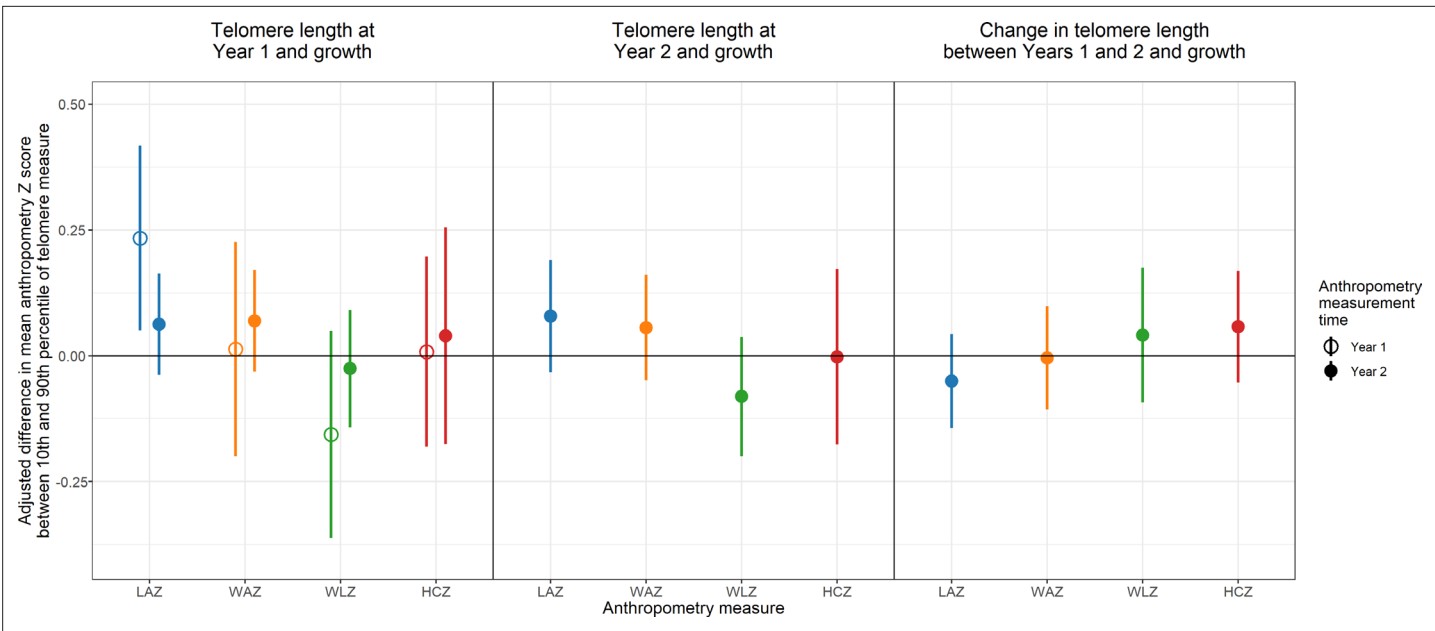

**Figure 2.** Adjusted association between telomere length and growth. Adjusted differences in mean anthropometry Z score between 10th and 90th percentile of telomere measure. LAZ = length-for-age Z score; WAZ = weight-for-age Z score; WLZ = weight-for-length Z score; HCZ = head circumference-for-age Z score.

The online version of this article includes the following source data, source code, and figure supplement(s) for figure 2:

**Source code 1.** Source code for *Figure 2*.

**Source data 1.** Source code for *Figure 2*.

**Figure supplement 1.** Association between telomere length at Year 1 and concurrent and subsequent growth.

**Figure supplement 1—source data 1.** Source data for *Figure 2—figure supplements 1–3*.

**Figure supplement 1—source code 1.** Source code for *Figure 2—figure supplements 1–3*.

**Figure supplement 2.** Association between telomere length at Year 2 and concurrent growth.

**Figure supplement 3.** Association between change in telomere length between Years 1 and 2 and growth.

*supplement 2*). There was also no association between TL at Year 2 and concurrent weight-for-age or head circumference-for-age Z score.

## Change in TL between Years 1 and 2 and growth

The median change in TL between Years 1 and 2 was 0.04 T/S ratio (IQR: –0.22 to 0.25) (*Table 1*). The median months between TL measurements at Years 1 and 2 was 13.9 months (IQR: 13.5–14.7). There was no evidence that changes in TL between Years 1 and 2 were associated with change in growth Z scores from Years 1 to 2, growth Z scores at Year 2, or growth velocity over the 1 year period (*Table 4* and *Supplementary file 1d* and *Figure 2*, *Figure 2—figure supplement 3*). We tested for the regression to the mean (RTM) effect by assessing the association between baseline TL at Year 1 and the change in TL between Years 1 and 2 (r = –0.18, p-value<0.001) (*Figure 3*). Because we observed that the correlation between Year 1 TL and TL change between Years 1 and 2 was partly due to regression to the mean (RTM) (*Figure 3*; *Berry et al., 1984*), we used the equation in Verhulst et al. to correct for the RTM effect in the primary analyses (*Verhulst et al., 2013*). Comparing associations between change in TL, uncorrected and corrected for the RTM effect, and growth outcomes yielded similar estimates (*Figure 3—figure supplements 1–2*).

## Post hoc analyses

Because the causal direction is not known for TL and growth, in post hoc analyses, we also examined potential associations in the opposite direction with growth as the exposure and TL as the outcome (*Supplementary file 1e–1h*). Higher weight-for-length Z score at Year 1 was associated with shorter TL

**Table 3.** Association between telomere length at Year 2 and growth.

| Exposure | Outcome | N | 10th percentile | 90th percentile | Outcome, 90th percentile vs. 10th percentile | | | |
|---|---|---|---|---|---|---|---|---|
| | | | | | Adjusted* | | | |
| | | | | | Predicted outcome at 10th percentile | Predicted outcome at 90th percentile | Coefficient (95% CI) | p-value |
| Telomere length at Year 2 (T/S ratio) | LAZ Year 2 | 592 | 1.1 | 1.7 | −1.68 | −1.6 | 0.08 (−0.03, 0.19) | 0.17 |
| | WAZ Year 2 | 611 | 1.1 | 1.7 | −1.84 | −1.8 | 0.06 (−0.05, 0.16) | 0.3 |
| | WLZ Year 2 | 615 | 1.1 | 1.7 | −0.98 | −1.1 | −0.08 (−0.2, 0.04) | 0.18 |
| | HCZ Year 2 | 612 | 1.1 | 1.7 | −1.94 | −1.9 | 0 (−0.18, 0.17) | 0.98 |

N, 10th percentile, and 90th percentile are from the adjusted analyses.

T/S ratio = unit for relative telomere length; LAZ = length-for-age Z score; WAZ = weight-for-age Z score; WLZ = weight-for-length Z score; HCZ = head circumference-for-age Z score.

*Adjusted for pre-specified covariates: Child age, child sex, birth order, prior child length and weight measurements from Year 1, season of measurement, caregiver-reported diarrhoea, mother's age, mother's height, mother's education level, mother's Center for Epidemiologic Studies Depression Scale Revised (CESD-R) score, mother's Perceived Stress Scale score, mother's lifetime exposure to physical, sexual, and emotional intimate partner violence, household food insecurity, number of children <18 years in the household, number of individuals living in the compound, distance in minutes to the primary water source, household floor materials, household wall materials, household electricity, and household assets (wardrobe, table, chair, clock, khat, chouki, radio, television, refrigerator, bicycle, motorcycle, sewing machine, mobile phone, cattle, goats, and chickens), and treatment arm (control or N + WSH) (*Supplementary file 1a*).

at Year 2 (−0.06 difference in T/S ratio [95% CI −0.11, −0.01] between the 90th and the 10th percentile [*Supplementary file 1f*]). This association was not significant after adjustment for multiple testing (FDR-corrected p-value≥0.05).

## Discussion

Our findings suggest that TL is associated with linear growth in the first year of life. Notably, longer relative TL was strongly associated with better linear growth at age 1 year. By age 2, this concurrent association was largely attenuated. These findings extend our inference of early childhood telomere dynamics within the context of the drinking water, sanitation, handwashing, and nutrition trial among young children in rural Bangladesh and support an adaptive role for telomere attrition, consistent with recently proposed hypotheses in evolutionary biology: the *costly maintenance hypothesis* and the *metabolic telomere attrition hypothesis* (*Casagrande and Hau, 2019*; *Young, 2018*).

The core tenet of the *costly maintenance hypothesis* is that there is an energetic cost to maintain long telomeres either through the active prevention of telomere attrition or promotion of telomere elongation (*Young, 2018*). Expending energy to sustain long telomeres limits energy resources necessary for other developmental or maintenance processes. The *metabolic telomere attrition hypothesis* expands on the *costly maintenance hypothesis* by proposing that the body will engage in active TL regulation as a means to address environmentally induced 'emergency states' that require increased energy expenditure (e.g., psychological stress, accelerated growth, nutrient shortage) (*Casagrande and Hau, 2019*). This short-term process prioritises resolution of the 'emergency state' to ensure survival over other bodily processes with potentially longer-term benefits. For example, a byproduct of telomere attrition is the acquisition of easily accessibly nucleotides for the body to redirect towards processes essential to address 'emergency states'.

In our previous study, children receiving the drinking water, sanitation, handwashing, and nutritional intervention had better growth and shorter TL at Year 1 compared to children in the control

**Table 4.** Association between change in telomere length and growth.

| Exposure | Outcome | N | 10th percentile | 90th percentile | Outcome, 90th percentile vs. 10th percentile | | | |
|---|---|---|---|---|---|---|---|---|
| | | | | | Adjusted* | | | |
| | | | | | Predicted outcome at 10th percentile | Predicted outcome at 90th percentile | Coefficient (95% CI) | p-value |
| Change in telomere length between Year 1 and Year 2 (T/S ratio) | LAZ Year 2 | 523 | −0.43 | 0.44 | −1.47 | −1.50 | −0.03 (−0.13, 0.07) | 0.62 |
| | WAZ Year 2 | 541 | −0.43 | 0.44 | −1.72 | −1.76 | −0.03 (−0.13, 0.07) | 0.53 |
| | WLZ Year 2 | 545 | −0.43 | 0.44 | −0.98 | −1.05 | −0.07 (−0.18, 0.04) | 0.24 |
| | HCZ Year 2 | 543 | −0.43 | 0.44 | −1.83 | −1.90 | −0.08 (−0.24, 0.09) | 0.36 |
| | Change in LAZ between Year 1 and Year 2 | 545 | −0.43 | 0.44 | −0.23 | −0.23 | 0 (−0.09, 0.08) | 1 |
| | Change in WAZ between Year 1 and Year 2 | 547 | −0.43 | 0.44 | −0.36 | −0.40 | −0.04 (−0.13, 0.05) | 0.36 |
| | Change in WLZ between Year 1 and Year 2 | 543 | −0.43 | 0.44 | −0.23 | −0.27 | −0.04 (−0.16, 0.08) | 0.53 |
| | Change in HCZ between Year 1 and Year 2 | 525 | −0.43 | 0.44 | −0.14 | −0.19 | −0.05 (−0.15, 0.06) | 0.39 |
| | Length velocity between Year 1 and Year 2 | 522 | −0.43 | 0.44 | 0.84 | 0.82 | −0.01 (−0.03, 0.01) | 0.29 |
| | Weight velocity between Year 1 and Year 2 | 522 | −0.43 | 0.44 | 0.15 | 0.15 | 0 (−0.01, 0.01) | 0.48 |
| | Head circumference velocity between Year 1 and Year 2 | 525 | −0.43 | 0.44 | 0.16 | 0.15 | −0.01 (−0.02, 0.01) | 0.35 |

N, 10th percentile, and 90th percentile are from the adjusted analyses.

T/S ratio = unit for relative telomere length; LAZ = length-for-age Z score; WAZ = weight-for-age Z score; WLZ = weight-for-length Z score; HCZ = head circumference-for-age Z score.

*Adjusted for pre-specified covariates: Child age, child sex, birth order, prior child length and weight measurements from Year 1 (included in Year 2 outcomes only), season of measurement, time between anthropometry measurements (included in growth velocity and change in growth measurements between Year 1 and Year 2 outcomes only), caregiver-reported diarrhoea, mother's age, mother's height, mother's education level, mother's Center for Epidemiologic Studies Depression Scale Revised (CESD-R) score, mother's Perceived Stress Scale score, mother's lifetime exposure to physical, sexual, and emotional intimate partner violence, household food insecurity, number of children <18 years in the household, number of individuals living in the compound, distance in minutes to the primary water source, household floor materials, household wall materials, household electricity, and household assets (wardrobe, table, chair, clock, khat, chouki, radio, television, refrigerator, bicycle, motorcycle, sewing machine, mobile phone, cattle, goats, and chickens), and treatment arm (control or N + WSH) (***Supplementary file 1a***).

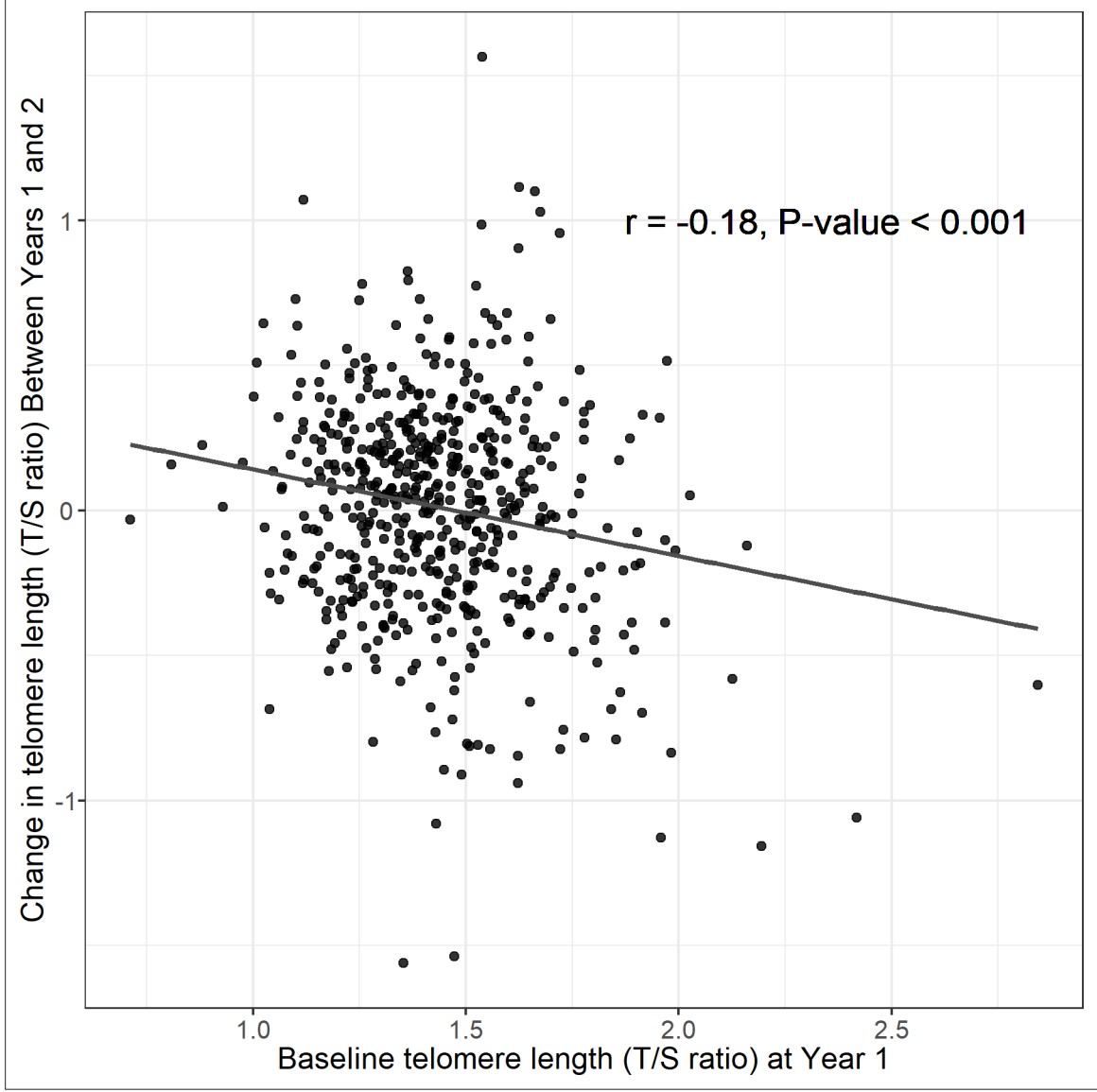

**Figure 3.** Regression to the mean assessment: association between telomere length at Year 1 and change in telomere length between Years 1 and 2. T/S ratio = unit for relative telomere length.

The online version of this article includes the following source code and figure supplement(s) for figure 3:

**Source code 1.** Source code for *Figure 3*.

**Figure supplement 1.** Regression to the mean comparison of unadjusted association between change in telomere length and growth.

**Figure supplement 1—source data 1.** Source data for *Figure 3—figure supplements 1 and 2*.

**Figure supplement 1—source code 1.** Source code for *Figure 3—figure supplements 1 and 2*.

**Figure supplement 2.** Regression to the mean comparison of adjusted association between change in telomere length and growth.

group (*Lin et al., 2017*; *Luby et al., 2018*). Linear growth velocity peaks during the first year of life and declines thereafter until puberty (*Avila, 2021*). Because early life growth involves increases in cell size and/or high rates of cell replication (*Monaghan and Ozanne, 2018*), during periods of rapid growth, energetic imbalances occur as the body expends increased energy to support cell replication processes (*Casagrande and Hau, 2019*). In the trial, the interventions created an optimal environment to promote child growth (*Luby et al., 2018*), but according to the *metabolic telomere attrition hypothesis*, rapid growth may have incurred a temporary energy debt which the body perceived as an 'emergency state' and actively shortened telomeres to resolve (*Casagrande and Hau, 2019*). Because

longer telomeres are energetically costly to maintain, according to the *costly maintenance hypothesis*, short-term accelerated telomere attrition would enable the redirection of energy towards processes involved in growth (*Young, 2018*) and would therefore be beneficial as a long-term strategy for health and longevity.

The findings of this study are consistent with the evolutionary framework and the previous trial results. Rapid telomere attrition may be essential to balance the energy deficit during short-term 'emergency states' to ensure individual survival or promote growth during the first year of life. Conversely, with age, perpetual telomere attrition over longer periods would result in crossing the critical lower threshold of TL, which would trigger cell senescence or apoptosis (*Casagrande and Hau, 2019*). Several studies in humans and other species suggest an association between longer TL, higher lifetime fitness, and increased longevity (*Young, 2018*). As a long-term strategy, long TL could represent a healthy initial setting that would confer resilience to cells, as they would be able to withstand more TL erosion events over the life course (*Casagrande and Hau, 2019*).

One caveat to note is that the concurrent associations lack the temporal ordering necessary to infer causal relationships. However, these results are consistent with other studies that have found positive associations between growth and TL: a study in Bangladesh found that low birth weight was associated with shorter TL at age 5 years (*Raqib et al., 2007*), and better early-life linear growth was associated with longer TL at age 21 years in a Filipino cohort (*Masterson et al., 2019*). Better growth during childhood is associated with reductions in adult mortality (*Ong et al., 2013*). Besides the *costly maintenance hypothesis* and the *metabolic telomere attrition hypothesis*, another potential mechanism underpinning the positive association between TL and growth is that infections contribute to poor growth (*Humphrey, 2009*) via increased immune activation leading to T-cell proliferation and accelerated telomere attrition (*Aviv, 2004*). Because the hypothalamic–pituitary–adrenal axis, oxidative stress, and immune activation affect TL and growth (*Casagrande and Hau, 2019*), forthcoming biochemical assessments of these systems within this cohort may further elucidate the biological pathways between TL and growth.

This is the first study to assess and demonstrate an association between TL and growth during the first 2 years of life. A limitation of the study is the lack of TL measurements prior to Year 1. Although a positive association was observed between simultaneous measurements of TL and linear growth at Year 1, we are unable to infer the temporal ordering of TL or growth on the causal pathway at Year 1. The study had limited statistical power to detect small differences in length-for-age, weight-for-age, weight-for-length, and head circumference-for-age Z scores and to delineate dose–response relationships between TL and growth; to address these issues, future studies should enrol sufficiently large sample sizes. Findings from this low-income, rural setting in Bangladesh, where growth failure is common, may not generalise to other populations; however, this well-characterised paediatric population provided a relevant setting to test the hypothesis. Although the heritability of TL is high (*Broer et al., 2013*), we did not assess maternal and paternal TL. However, the potential bias from this source is minimal because parental TL does not satisfy the criteria for a potential confounder: although parental TL is strongly associated with child TL, there is no evidence to suggest that parental TL is associated with child growth. To address genetics, maternal height was screened as a covariate for inclusion in adjusted models. The study also screened several other covariates thought to be associated with TL and growth, but due to the observational nature of this assessment, the potential for unmeasured confounding remains a limitation.

Because rates of TL attrition in somatic tissue reflect stem cell replication and the study only assessed TL in whole blood, measurements in other minimally proliferative tissue types might yield differences in TL (*Daniali et al., 2013*). However, within the same individual, some studies suggest high synchrony between TL attrition rates in peripheral blood and those within other somatic tissues (*Daniali et al., 2013*; *Takubo et al., 2002*). Whole blood TL represents a composite measurement of TL from a heterogeneous population of cell types in varying abundance. Although change in TL is correlated across peripheral blood mononuclear cells, B cells, and T cells within the same individual, TL shortens faster in peripheral blood mononuclear cells compared to whole blood cells and B cells compared to T cells (*Lin et al., 2016*; *Zole and Ranka, 2019*). These differences in TL attrition may affect individual results across similar studies assessing TL in distinct immune cell subsets. Although terminal restriction fragment analysis via the Southern blot procedure is considered the 'gold standard' to measure absolute TL, the assay requires a prohibitively large volume of blood (*Lai et al.,*

*2018*). Therefore, we adopted the quantitative polymerase chain reaction (qPCR) approach, which requires less DNA, measures relative TL by determining the ratio of telomeric DNA (T) to a reference single-copy gene signal (S) (*Cawthon, 2002*; *Lin et al., 2010*), and is a validated approach that is widely used in epidemiological settings. TLs measured by Southern blot and qPCR are strongly correlated ($r > 0.9$) (*Aviv et al., 2011*).

The first 2 years of life represent a sensitive period for child growth and development. Although TL was associated with linear growth during the first year of life, this study highlights our limited understanding of the underlying biological mechanisms along these pathways. Telomeres may play a causal adaptive role in child growth or serve as a 'molecular clock' that gauges cumulative environmental exposures that affect both early-life telomere dynamics and growth (*Casagrande and Hau, 2019*; *Shalev et al., 2013*; *Young, 2018*). A large body of existing research links stress to short TL among all ages; however, the unexpected findings from this study suggest that TL dynamics reflect different processes during childhood and have implications for healthy trajectories. Prospective birth cohort studies conducted in a range of geographical contexts, with frequent longitudinal measurements, would further delineate the relationship between early life telomeres and growth.

## Acknowledgements

We thank the families who participated in the WASH Benefits study and the incredible icddr,b staff for their valuable contributions. This work was supported by Global Development grant [OPPGD759] from the Bill & Melinda Gates Foundation to the University of California, Berkeley and by the National Institute of Allergy and Infectious Diseases of the National Institutes of Health [grant number K01AI136885 to AL]. The content is solely the responsibility of the authors and does not necessarily represent the official views of the National Institutes of Health. icddr,b is grateful to the Governments of Bangladesh, Canada, Sweden, and the United Kingdom for providing core/unrestricted support.

## Additional information

### Competing interests

Jue Lin: is a co-founder of Telomere Diagnostics Inc, a telomere measurement company. Assays and all other activity for the current report are, however, unrelated to this company.. The other authors declare that no competing interests exist.

### Funding

| Funder | Grant reference number | Author |
| --- | --- | --- |
| Bill and Melinda Gates Foundation | OPPGD759 | John M Colford |
| National Institute of Allergy and Infectious Diseases | K01AI136885 | Audrie Lin |

The funders had no role in study design, data collection and interpretation, or the decision to submit the work for publication.

### Author contributions

Audrie Lin, Conceptualization, Data curation, Formal analysis, Funding acquisition, Investigation, Methodology, Project administration, Resources, Supervision, Writing – original draft, Writing – review and editing; Andrew N Mertens, Data curation, Formal analysis, Investigation, Methodology, Project administration, Resources, Software, Supervision, Validation, Visualization, Writing – review and editing; Benjamin F Arnold, Data curation, Funding acquisition, Methodology, Resources, Writing – review and editing; Sophia Tan, Visualization, Writing – review and editing; Jue Lin, Methodology, Supervision, Writing – review and editing; Christine P Stewart, Alan E Hubbard, Ruchira Tabassum Naved, Lia CH Fernald, Funding acquisition, Methodology, Supervision, Writing – review and editing; Shahjahan Ali, Investigation, Supervision, Writing – review and editing; Jade Benjamin-Chung, Funding acquisition, Project administration, Writing – review and editing; Abul K Shoab, Data

curation, Investigation, Writing – review and editing; Md Ziaur Rahman, Syeda L Famida, Md Saheen Hossen, Palash Mutsuddi, Salma Akther, Data curation, Investigation, Supervision, Writing – review and editing; Mahbubur Rahman, Leanne Unicomb, Funding acquisition, Project administration, Supervision, Writing – review and editing; Md Mahfuz Al Mamun, Kausar Parvin, Funding acquisition, Investigation, Writing – review and editing; Firdaus S Dhabhar, Patricia Kariger, Funding acquisition, Writing – review and editing; Stephen P Luby, John M Colford, Funding acquisition, Project administration, Resources, Supervision, Writing – review and editing

### Author ORCIDs
Audrie Lin http://orcid.org/0000-0002-3877-3469
Andrew N Mertens http://orcid.org/0000-0002-1050-6721
Benjamin F Arnold http://orcid.org/0000-0001-6105-7295
Shahjahan Ali http://orcid.org/0000-0003-3883-1208

### Ethics
Human subjects: Clinical trial registration: The trial was registered at ClinicalTrials.gov (NCT01590095). Human subjects: Primary caregivers of all children provided written informed consent. The study protocols were approved by human subjects committees at icddr,b (PR-11063 and PR-14108), the University of California, Berkeley (2011-09-3652 and 2014-07-6561) and Stanford University (25863 and 35583).

### Decision letter and Author response
Decision letter https://doi.org/10.7554/eLife.60389.sa1

---

## Additional files

### Supplementary files
• Supplementary file 1. Supplementary tables. (a) Pre-specified covariates screened for inclusion in fully adjusted models. (b) Association Between Telomere Length at Year 1 and Growth. (c) Association Between Telomere Length at Year 2 and Growth. (d) Association Between Change in Telomere Length and Growth. (e) Post-hoc Analyses: Association Between Growth at Month 3 and Subsequent Telomere Length. (f) Post-hoc Analyses: Association Between Growth at Year 1 and Subsequent Telomere Length. (g) Post-hoc Analyses: Association Between Change in Growth and Telomere Length. (h) Post-hoc Analyses: Association Between Growth Velocity and Telomere Length.

• Transparent reporting form

• Source code 1. Source code for *Tables 2–4* and *Supplementary file 1b–1d*.

• Source code 2. Source code for *Supplementary file 1e–1h*.

• Source data 1. Source data for *Supplementary file 1e–1h*.

• Source data 2. Source data for *Tables 2–4* and *Supplementary file 1b–1d*.

### Data availability
The WASH Benefits data and code that support the findings of this study are available in Open Science Framework (https://osf.io/9snat/).

The following dataset was generated:

| Author(s) | Year | Dataset title | Dataset URL | Database and Identifier |
|---|---|---|---|---|
| Lin A, Mertens AN, Tan S | 2021 | WASH Benefits Bangladesh Analysis of Telomere and Growth Outcomes | https://osf.io/9snat/ | Open Science Framework, 9snat |

The following previously published datasets were used:

| Author(s) | Year | Dataset title | Dataset URL | Database and Identifier |
|---|---|---|---|---|
| Lin A, Mertens AN, Arnold BF | 2017 | WASH Benefits Bangladesh Analysis of Telomere Outcomes | https://osf.io/evc98/ | Open Science Framework, 10.17605/OSF.IO/EVC98 |

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
