## [Decision Letter]

**Acceptance summary:**

Following up on a previous *eLife* paper in which they demonstrated that water, sanitation, handwashing, and nutritional intervention improved linear growth and unexpectedly shortened childhood telomere length (TL), Lin et al., now show that TL is associated with concurrent growth. Although existing research links stress to TL attrition, their unexpected findings suggest a more complex relationship between childhood TL dynamics and healthy trajectories.

**Decision letter after peer review:**

Thank you for submitting your article "Telomere length is associated with subsequent growth in children in rural Bangladesh" for consideration by *eLife*. Your article has been reviewed by 3 peer reviewers, and the evaluation has been overseen by Eduardo Franco as the Senior and Reviewing Editor. The following individual involved in review of your submission has agreed to reveal their identity: Janet Wojcicki (Reviewer #3).

As is customary in *eLife*, the reviewers have discussed their critiques with one another. What follows below is my lightly edited compilation of the essential and ancillary points provided by reviewers in their critiques and in their interaction post-review (first person statements are kept for nuancing the comments). Please submit a revised version that addresses these concerns directly. Although we expect that you will address these comments in your response letter, we also need to see the corresponding revision in the text of the manuscript. Some of the reviewers' comments may seem to be simple queries or challenges that do not prompt revisions to the text. Please keep in mind, however, that readers may have the same perspective as the reviewers. Therefore, it is essential that you attempt to amend or expand the text to clarify the narrative accordingly.

Summary:

Following up on a previous *eLife* paper in which they demonstrated that water, sanitation, handwashing, and nutritional intervention improved linear growth and unexpectedly shortened childhood telomere length (TL), Lin et al., now show that TL is associated with concurrent and subsequent growth. Although existing research links stress to TL attrition, our unexpected findings suggest a more complex relationship between childhood TL dynamics and healthy trajectories.

Essential revisions:

Although the idea of the paper is interesting, I find that the main conclusions of the authors are not supported by their statistical analyses (lack of multiple testing correction). My main concern with the study relates to the statistical power and the results reported as significant. The sample size is small and therefore most of the associations tested were not significant. The associations reported as significant showed p-values between 1% and 5%. The number of tests performed in the study are quite large, and some multiple correction testing should be included, which would probably deem most of them as non-significant.

I would also like to see the analyses performed on TL as a continuous variable instead of quartiles of its distribution. The authors hypothesise that TL has an effect on the studied traits, and I think this is expected to be linear, however, this is not tested explicitly. The tests comparing quartiles of TL distribution are somewhat arbitrary (just Q1 v. all others) and although the means are nominally significantly different in some cases, they go up and down rather than showing a linear pattern: From Table 2, LAZ year 1. Q1mean: -1.66 Q2mean: -1.33 Q3mean: -1.44 Q4mean: -1.27. The reason behind this analysis is not justified as the lack of linearity is not discussed in the results. I think the associations claimed by the authors are not supported by their analyses.

Some of the tables are somewhat difficult to navigate because they contain a lot of information, but this is not particularly problematic.

In your limitations section, you should comment on whether your TL measurement approach was the most optimal. Some would argue that Southern blot is the better than PCR.

My understanding is that the heritability of telomere length is quite large (~70%, see e.g., https://www.nature.com/articles/ejhg2012303) As no genetic data is available this cannot be addressed easily but I would like to see a bit of discussion with respect of the implications of this. Since it is not mentioned I assume all the children are unrelated to each other, otherwise results could be biased because of the genetic structure of the population.

Reporting results without correcting for the relevant covariates is misleading and unnecessarily complicates the main tables.

The authors indicate that median TL in year 1 is 14 months and in year 2, 28 months but there are no additional data given including SD or mean, range.

Change in TL is a mean of 0.04 but again there is no data given on mean time period that change is measured and what are the ranges for children? Age is adjusted for in analyses (as days) but there is no descriptive analysis of age in relation to TL measurement or TL change. Also, the authors indicate that there is an adjustment for regression to the mean due to the correlation between timepoint 1 and change in TL but also no discussion of how the authors discerned there was regression to the mean. Also how did this adjustment change findings?

The discussion is cursory in regards to paradoxical findings regarding accelerated TL attrition and decreased HC as well as shorter TL at year 1 and lower WLZ at year 2. The authors only conclude that growth is more complex that previously indicated without providing any hypotheses of what could explain this? The authors should review other publications that have found paradoxical results in TL with metabolic change including the review paper by Casagrande and Hau on Telomere attrition (Biology Letters 2019).

Lines 80-81 – should state the intervention was associated with shortened TL – not that it itself shortened TL

Lines 111-112 – the abbreviations were not previously defined. Please define these.

---

## [Author Response]

Essential revisions:Although the idea of the paper is interesting, I find that the main conclusions of the authors are not supported by their statistical analyses (lack of multiple testing correction). My main concern with the study relates to the statistical power and the results reported as significant. The sample size is small and therefore most of the associations tested were not significant. The associations reported as significant showed p-values between 1% and 5%. The number of tests performed in the study are quite large, and some multiple correction testing should be included, which would probably deem most of them as non-significant.

In the original submission, we had not corrected for multiple testing due to the nature of the exploratory investigation. However, we agree with the Reviewer’s suggestion to include multiple testing for the reasons outlined. We have revised the manuscript and have added a footnote to each table and asterisks to p-values indicating “^*^P-value < 0.05 after adjusting for multiple comparisons using the Benjamini-Hochberg procedure”. We have also added a sentence to the Methods section: “We reported unadjusted p-values and adjusted for multiple testing by controlling the false discovery rate (at an FDR of 5%) within each hypothesis using the Benjamini-Hochberg procedure.” The key finding with a p-value < 0.05 after adjusting for multiple comparisons using the Benjamini-Hochberg procedure was as follows:

“At Year 1, longer TL was concurrently associated with taller children (+0.23 SD adjusted difference in length-for-age Z score between the 10th and 90th percentile (95% CI 0.05, 0.42; FDR-corrected p-value = 0.01)) (Table 2 and Supplementary file 1b; Figure 2 and Figure 2—figure supplement 1).”

I would also like to see the analyses performed on TL as a continuous variable instead of quartiles of its distribution. The authors hypothesise that TL has an effect on the studied traits, and I think this is expected to be linear, however, this is not tested explicitly. The tests comparing quartiles of TL distribution are somewhat arbitrary (just Q1 v. all others) and although the means are nominally significantly different in some cases, they go up and down rather than showing a linear pattern: From Table 2, LAZ year 1. Q1mean: -1.66 Q2mean: -1.33 Q3mean: -1.44 Q4mean: -1.27. The reason behind this analysis is not justified as the lack of linearity is not discussed in the results. I think the associations claimed by the authors are not supported by their analyses.

In accordance with best practices to promote transparency and reproducibility, the original analyses performed on quartiles of TL followed the pre-specified and registered analysis plan posted on Open Science Framework (https://osf.io/8d6xf/). We have addressed the Reviewer’s concerns by changing the analytical approach and performing the analyses on telomere length as a continuous variable instead of quartiles of its distribution. We made the following revisions in light of our revised analytical approach:

– We have revised the Methods section:

“We summarized mean anthropometric Z scores [length-for-age, weight-for-age, weight-for-length, and head circumference-for-age Z scores], change in anthropometric Z scores, length velocity, weight velocity, and head circumference velocity across the distribution of TL or change in TL using natural smoothing splines (generalized additive models), both unadjusted and adjusted for potential confounding covariates. We estimated differences in Z scores and pointwise confidence intervals compared to a reference level of the lowest observed TL or change in TL. We reported the differences and confidence intervals between the 10th percentile and 90th percentile of TL or change in TL distribution using predictions from the generalized additive models.”

– All Tables, Figures, and Supplementary Materials have been revised to reflect this change in analytical approach.

– We have also extensively revised the Title, Abstract, Results, and Discussion sections to reflect the changed results and inference. The key finding with a p-value < 0.05 after adjusting for multiple comparisons using the Benjamini-Hochberg procedure was as follows:

“At Year 1, longer TL was concurrently associated with taller children (+0.23 SD adjusted difference in length-for-age Z score between the 10th and 90th percentile (95% CI 0.05, 0.42; FDR-corrected p-value = 0.01)) (Table 2 and Supplementary file 1b; Figure 2 and Figure 2—figure supplement 1).”

Some of the tables are somewhat difficult to navigate because they contain a lot of information, but this is not particularly problematic.

Thank you for this recommendation. We have simplified the Main Tables formatting by moving the unadjusted estimates to Supplementary Tables to improve readability.

In your limitations section, you should comment on whether your TL measurement approach was the most optimal. Some would argue that Southern blot is the better than PCR.

Thank you for this suggestion. We have included the following language in the Limitations section of the Discussion:

“Although terminal restriction fragment analysis via the Southern blot procedure is considered the ‘gold standard’ to measure absolute telomere length, the assay requires a prohibitively large volume of blood (Lai, Wright, and Shay, 2018). Therefore, we adopted the quantitative polymerase chain reaction (qPCR) approach, which requires less DNA, measures relative telomere length by determining the ratio of telomeric DNA (T) to a reference single-copy gene signal (S) (Cawthon, 2002; J. Lin et al., 2010), and is a validated approach that is widely used in epidemiological settings. Telomere lengths measured by Southern blot and qPCR are strongly correlated (r > 0.9) (Aviv et al., 2011).”

My understanding is that the heritability of telomere length is quite large (~70%, see e.g., https://www.nature.com/articles/ejhg2012303) As no genetic data is available this cannot be addressed easily but I would like to see a bit of discussion with respect of the implications of this. Since it is not mentioned I assume all the children are unrelated to each other, otherwise results could be biased because of the genetic structure of the population.

We thank the Reviewer for highlighting this issue. We have added the following language to the limitation section of the Discussion and included this citation:

“Although the heritability of TL is high (Broer et al., 2013), we did not assess maternal and paternal telomere length. However, the potential bias from this source is minimal because parental TL does not satisfy the criteria for a potential confounder: although parental TL is strongly associated with child TL, there is no evidence to suggest that parental TL is associated with child growth. To address genetics, maternal height was screened as a covariate for inclusion in adjusted models.”

Our analysis included three sets of twins. We performed a sensitivity analysis excluding one twin from each pair, which yielded similar results as our primary analysis (compare effect estimates in Author response table 1 to Table 2). Thus, including the three sets of twins in our primary analysis resulted in minimal bias.

**Author response table 1. sa2table1:** Twin Sensitivity Analysis: Association Between Telomere Length at Year 1 and Growth.

Exposure	Outcome	N	10th Percentile	90th Percentile	Outcome, 90th Percentile v. 10th Percentile			
					Adjusted			
					Predicted Outcome at 10th Percentile	Predicted Outcome at 90th Percentile	Coefficient (95% CI)	P-value
Telomere length at Year 1	LAZ Year 1	635	1.2	1.7	-1.52	-1.32	0.2 (0.02, 0.37)	0.03*
	WAZ Year 1	634	1.2	1.7	-1.18	-1.20	-0.02 (-0.22, 0.19)	0.89
	WLZ Year 1	633	1.2	1.7	-0.66	-0.83	-0.17 (-0.37, 0.03)	0.10
	HCZ Year 1	635	1.2	1.7	-1.72	-1.73	-0.01 (-0.2, 0.18)	0.92
	LAZ Year 2	540	1.2	1.7	-1.50	-1.44	0.06 (-0.04, 0.16)	0.21
	WAZ Year 2	540	1.2	1.7	-1.74	-1.69	0.05 (-0.05, 0.16)	0.35
	WLZ Year 2	566	1.2	1.7	-0.97	-1.00	-0.03 (-0.15, 0.08)	0.58
	HCZ Year 2	563	1.2	1.7	-1.87	-1.82	0.04 (-0.13, 0.21)	0.65
	Change in LAZ between Year 1 and Year 2	566	1.2	1.7	-0.24	-0.29	-0.05 (-0.14, 0.05)	0.35
	Change in WAZ between Year 1 and Year 2	570	1.2	1.7	-0.37	-0.38	-0.01 (-0.11, 0.09)	0.87
	Change in WLZ between Year 1 and Year 2	563	1.2	1.7	-0.25	-0.22	0.03 (-0.1, 0.17)	0.66
	Change in HCZ between Year 1 and Year 2	543	1.2	1.7	-0.16	-0.11	0.06 (-0.05, 0.17)	0.32
	Length velocity between Year 1 and Year 2	539	1.2	1.7	0.80	0.80	0 (-0.02, 0.03)	0.76
	Weight velocity between Year 1 and Year 2	539	1.2	1.7	0.15	0.15	0 (-0.01, 0.01)	0.84
	Head circumference velocity between Year 1 and Year 2	543	1.2	1.7	0.15	0.16	0.01 (0, 0.02)	0.14

N, 10th Percentile, and 90th Percentile are from the adjusted analyses.

P-value < 0.05 after adjusting for multiple comparisons using the Benjamini-Hochberg procedure.

Reporting results without correcting for the relevant covariates is misleading and unnecessarily complicates the main tables.

We have simplified the Main Tables by removing the unadjusted results.

The authors indicate that median TL in year 1 is 14 months and in year 2, 28 months but there are no additional data given including SD or mean, range.Change in TL is a mean of 0.04 but again there is no data given on mean time period that change is measured and what are the ranges for children? Age is adjusted for in analyses (as days) but there is no descriptive analysis of age in relation to TL measurement or TL change. Also, the authors indicate that there is an adjustment for regression to the mean due to the correlation between timepoint 1 and change in TL but also no discussion of how the authors discerned there was regression to the mean. Also how did this adjustment change findings?

We thank the Reviewer for these comments. The median and interquartile range (IQR) for TL at Year 1, TL at Year 2, and change in TL between Years 1 and 2 were included in Table 1. We have added the median and IQR for child age in months at Year 1 and Year 2 to Table 1. We have also added months between telomere length measurements at Year 1 and Year 2 to Table 1 to address the median time period and IQR that change in telomere length was measured. We have added the following lines to the Results section:

– “The median age of the children was 14.3 (IQR: 12.6 to 15.6) months at Year 1 and 28.2 (IQR: 26.9 to 29.6) months at Year 2 (Table 1).”

– “The median months between telomere length measurements at Years 1 and 2 was 13.9 months (IQR: 13.5 to 14.7).”

We created one new Figure (Figure 3) and two new Figure supplements (Figure 3—figure supplements 1–2) (included below) to address the regression to the mean issue. Figure 3 shows an association (r = -0.18, p-value <0.001) between baseline telomere length at Year 1 and the change in telomere length between Years 1 and 2. Because there is a correlation, the regression to the mean effects is present. We corrected for the regression to the mean effect using the equation in Verhulst *et al.,* (Verhulst, Aviv, Benetos, Berenson, and Kark, 2013).

Figure 3—figure supplement 1 shows the regression to the mean comparison of unadjusted association between change in telomere length and growth outcomes (growth at Year 2, change in growth between Years 1 and 2, and growth velocity between Years 1 and 2). In the figure, we compared the analyses using the uncorrected change in telomere length versus the regression to the mean-corrected change in telomere length and displayed the comparison of the unadjusted differences in mean anthropometry Z-scores between the 10^th^ and 90^th^ percentile of telomere length measurement. Uncorrected and regression to the mean-corrected differences were similar.

Similarly, Figure 3—figure supplement 2 shows the same uncorrected versus regression to the mean-corrected comparison with the adjusted differences in mean anthropometry Z-scores between the 10^th^ and 90^th^ percentile of telomere length measurement. These analyses were adjusted for pre-specified covariates. The unadjusted and adjusted results were similar.

In addition to adding three figures to the Supplementary Tables, we also added the following paragraph to the Results section to summarize the regression to the mean assessment and these results:

“We tested for the regression to the mean (RTM) effect by assessing the association between baseline telomere length at Year 1 and the change in telomere length between Years 1 and 2 (r = -0.18, p-value < 0.001) (Figure 3). Because we observed that the correlation between Year 1 TL and TL change between Years 1 and 2 was partly due to regression to the mean (RTM) (Figure 3) (Berry, Eaton, Ekholm, and Fox, 1984), we used the equation in Verhulst et al. to correct for the RTM effect in the primary analyses (Verhulst et al., 2013). Comparing associations between change in telomere length, uncorrected and corrected for the RTM effect, and growth outcomes yielded similar estimates (Figure 3—figure supplements 1–2).”

The discussion is cursory in regards to paradoxical findings regarding accelerated TL attrition and decreased HC as well as shorter TL at year 1 and lower WLZ at year 2. The authors only conclude that growth is more complex that previously indicated without providing any hypotheses of what could explain this? The authors should review other publications that have found paradoxical results in TL with metabolic change including the review paper by Casagrande and Hau on Telomere attrition (Biology Letters 2019).

After addressing the Reviewers’ comments above: changing the analytical approach and performing the analyses on telomere length as a continuous variable instead of quartiles of its distribution and adjusting for multiple testing using the Benjamini-Hochberg procedure, the paradoxical results of decreased HCZ and lower WLZ were no longer significant. However, we thank the Reviewer for the suggestions to review the Casagrande and Hau paper. The Reviewer’s suggestion greatly strengthened the Discussion section. Based on the ideas outlined in the Casagrande and Hau review paper (the *costly maintenance hypothesis* and the *metabolic telomere attrition hypothesis*), we revised and added new paragraphs to the Discussion section:

“Our findings suggest that TL is associated with linear growth in the first year of life. Notably, longer relative TL was strongly associated with better linear growth at age 1 year. By age 2, this concurrent association was largely attenuated. These findings extend our inference of early childhood telomere dynamics within the context of the drinking water, sanitation, handwashing, and nutrition trial among young children in rural Bangladesh and support an adaptive role for telomere attrition, consistent with recently proposed hypotheses in evolutionary biology: the costly maintenance hypothesis and the metabolic telomere attrition hypothesis (Casagrande and Hau, 2019; Young, 2018).

The core tenet of the costly maintenance hypothesis is that there is an energetic cost to maintain long telomeres either through the active prevention of telomere attrition or promotion of telomere elongation (Young, 2018). Expending energy to sustain long telomeres limits energy resources necessary for other developmental or maintenance processes. The metabolic telomere attrition hypothesis expands on the costly maintenance hypothesis by proposing that the body will engage in active TL regulation as a means to address environmentally-induced ‘emergency states’ that require increased energy expenditure (e.g., psychological stress, accelerated growth, nutrient shortage) (Casagrande and Hau, 2019). This short-term process prioritizes resolution of the ‘emergency state’ to ensure survival over other bodily processes with potentially longer-term benefits. For example, a byproduct of telomere attrition is the acquisition of easily accessibly nucleotides for the body to redirect towards processes essential to address ‘emergency states’.

In our previous study, children receiving the drinking water, sanitation, handwashing, and nutritional intervention had better growth and shorter TL at Year 1 compared to children in the control group (A. Lin et al., 2017; Luby et al., 2018). Linear growth velocity peaks during the first year of life and declines thereafter until puberty (Avila, 2021). Because early life growth involves increases in cell size and / or high rates of cell replication (Monaghan and Ozanne, 2018), during periods of rapid growth, energetic imbalances occur as the body expends increased energy to support cell replication processes (Casagrande and Hau, 2019). In the trial, the interventions created an optimal environment to promote child growth (Luby et al., 2018), but according to the metabolic telomere attrition hypothesis, rapid growth may have incurred a temporary energy debt which the body perceived as an ‘emergency state’ and actively shortened telomeres to resolve (Casagrande and Hau, 2019). Because longer telomeres are energetically costly to maintain, according to the costly maintenance hypothesis, short-term accelerated telomere attrition would enable the redirection of energy towards processes involved in growth (Young, 2018) and would therefore be beneficial as a long-term strategy for health and longevity.

The findings of this study are consistent with the evolutionary framework and the previous trial results. Rapid telomere attrition may be essential to balance the energy deficit during short-term ‘emergency states’ to ensure individual survival or promote growth during the first year of life. Conversely, with age, perpetual telomere attrition over longer periods would result in crossing the critical lower threshold of TL, which would trigger cell senescence or apoptosis (Casagrande and Hau, 2019). Several studies in humans and other species suggest an association between longer telomere length, higher lifetime fitness, and increased longevity (Young, 2018). As a long-term strategy, long TL could represent a healthy initial setting that would confer resilience to cells, as they would be able to withstand more TL erosion events over the life course (Casagrande and Hau, 2019).”

Lines 80-81 – should state the intervention was associated with shortened TL – not that it itself shortened TL

We have revised this sentence:

“The interventions improved linear growth (length for age) but were unexpectedly associated with shortened TL during the first year of life (Lin et al., 2017; Luby et al., 2018), findings that challenged the prevailing paradigm that early-life stressors shorten TL (Ridout et al., 2015) and motivated the hypothesis of the present study: that accelerated TL attrition in early life could be associated with improved growth.”

Lines 111-112 – the abbreviations were not previously defined. Please define these.

We have revised the sentence to replace these abbreviations with the full descriptive nomenclature:

“The median length-for-age Z score, weight-for-age Z score, weight-for-length Z score, and head circumference-for-age Z score was -1.41, -1.30, -0.89, and -1.81 respectively (Table 1).”